# Identifying Strategies for Effective Biodiversity Preservation and Species Status of Chilean Amphibians

**DOI:** 10.3390/biology13030169

**Published:** 2024-03-05

**Authors:** Marcela A. Vidal, Nayadet Henríquez, Cristian Torres-Díaz, Gonzalo Collado, Ian S. Acuña-Rodríguez

**Affiliations:** 1Center for Ñuble Studies, Universidad del Bío-Bío, Av. Andrés Bello 720, Chillan 3800708, Chile; 2Biodiversity and Global Change Research Group, Facultad de Ciencias, Universidad del Bío-Bío, Av. Andrés Bello 720, Chillan 3800708, Chile; nayhenrim@gmail.com (N.H.); crtorres@ubiobio.cl (C.T.-D.); gcollado@ubiobio.cl (G.C.); 3Instituto de Investigación Interdisciplinaria (I3), Universidad de Talca, Campus Lircay, Avda, Lircay s/n, Talca 3465548, Chile; ian.acuna@utalca.cl

**Keywords:** amphibians, biodiversity, Chile, conservation, threats

## Abstract

**Simple Summary:**

In this study, we focus on the evaluation of the conservation priority of amphibian species in Chile. We emphasize the importance of establishing standardized criteria to evaluate and classify this priority, considering factors such as extinction risk, geographic distribution, and ecological importance. We establish priority categories for species, allowing us to identify those that require immediate attention in terms of conservation. Six species with a high priority, eight with a medium priority, twenty-two with a low priority, and twenty-two with no priority were identified. We emphasize the relevance of the Telmatobiidae and Alsodidae families as conservation priorities, highlighting the need to focus our conservation efforts on these amphibian families in Chile. This approach provides a solid basis for informed decision making in the allocation of limited resources for biodiversity conservation.

**Abstract:**

Resources are limited in global biodiversity conservation efforts, which emphasizes the significance of setting conservation priorities. Using standardized criteria, we evaluated 58 amphibian species in Chile to determine their conservation priority (CP). Species with insufficient historical data had their values marked as missing. With a median value of *p* = 1.67, the results demonstrated CP values ranging from *p* = 0.48 to *p* = 3.0, classifying species into priority and non-priority groups. Four levels were established for the priority categories: no priority, low priority, medium priority, and high priority. Additionally, the Telmatobiidae and Alsodidae families were identified as two more priority families. Notably, the species with the highest priority were found to be *T. halli*, *T. fronteriensis*, *T. philippii*, *T. chusmisensis*, *A. pehuenche*, and *Alsodes tumultuosus*, where *T. philippii* and *T. fronteriensis* have equal priority for conservation at the national level according to the conservation priority analysis. Eight priority families—the Alsodidae, Batrachylidae, Bufonidae, Ceratophryidae, Leptodactylidae, Rhinodermatidae, and Telmatobiidae—were determined, and 14 species—or 24% of the species examined—need further study. Based on the conservation priority analysis, the species *T. fronteriensis* and *T. philippii* share the highest priority for conservation at the national level (*p* = 2.50). With 70% of the amphibians under study being threatened mainly by habitat loss, pollution, and emerging diseases, the creation of conservation categories made the threat assessment process easier. Due to a lack of information on geographic distribution and abundance, quantitatively classifying amphibians in Chile remains difficult. The analysis of conservation priorities and potential extinction threats informs appropriate management strategies.

## 1. Introduction

The progressive growth of the world population and the unsustainable demand for natural resources directly threaten biodiversity [1]. According to Pimm et al. [2] and Zumbado et al. [3], there is a growing trend in the extinction of species as a result of habitat destruction and contamination, accelerated rates of climate change, excessive capture and removal of these animals from the wild, excessive use as food, and severe infectious diseases. In this context, if it is expected that the few remaining wilderness areas that support this biodiversity will be effectively protected, it is critical to evaluate the effects of this biotic erosion (i.e., biodiversity loss resulting from human intention in the world’s ecosystems) on various aspects of diversity and overall ecosystem properties [4,5,6].

The categorization of a species’ conservation status allows for appropriate planning and population management from a sustainable perspective [7]. Globally, the most significant efforts to determine the conservation status of species are led by the non-governmental organization IUCN, which publishes lists of globally threatened species (www.iucnredlist.org (accessed on 1 December 2023)). Based on these procedures, countries have adapted their species classification systems or adopted the conditions already established by the organization. According to the latest update from the Union for Conservation of Nature [8], approximately 28% of all globally described species are considered threatened, with amphibians, mammals, and conifers being the groups at the highest risk.

Vertebrates, which make up only 3% of all known animal species, are thought to be more complex in this regard. They are important for maintaining the stability of various ecosystems and are highly valued in most human communities worldwide on cultural, economic, and social levels [9] because a systemic framework known as ecosystem services conceptualizes the variety of interconnected values that ecosystems offer to humanity, many of which may be lost or deteriorated through exploitation limited to utilitarian purposes [10]. However, upon the examination of data from about 25,780 vertebrate species, comprising fish (a sample of >1200 species), reptiles (a sample of 1500 species), mammals (5498 species), birds (10,027 species), and amphibians (6638 species), it was discovered that approximately one-fifth of these species fall into the “endangered” category [8]. This ranges from 13% for birds to 41% for amphibians [11]. In fact, between 1980 and 2004, the assessments of the International Union for Conservation of Nature (IUCN) red list for the latter order—the most threatened among the vertebrate orders—were altered by 3.4%, or 0.14% annually. The data suggests that the population status of 662 amphibian species is declining, which has resulted in a higher percentage of these species being classified as “endangered” on the IUCN Red List [12,13]. Numerous Neotropical amphibian species included in these threat categories face declines and local extinctions brought on by invasive predators, emerging diseases, and livestock disturbance of habitat [14]. Additionally, the loss of these group’s evolutionary histories is a consequence of species erosion brought on by climate change [15].

Regionally, Chile has a distinct biodiversity legacy due to the heterogeneous distribution of species and endemism throughout the nation [16,17,18,19]. Of the 35,000 species that have been described, about 25% are endemic [20,21]. The Andes mountain range, which rises to 6000 m in the east, the Pacific Ocean in the west, Antarctica in the south, and the world’s driest desert in the north are the natural barriers that encircle Chile, contributing to its uniqueness [21,22,23]. Specifically, the central region of Chile—that is, between 25 and 47° S on the Pacific coast and the Andean peaks—has been widely identified as a “Chilean hotspot” [24]. Indeed, this region is included in various global hotspot lists as a highly relevant place for global diversity [21,22,23,24]. Consequently, it is characterized by a high concentration of endemic species with restricted distribution ranges and considerable vulnerability [25].

Amphibians, being ectothermic organisms that rely on freshwater for reproduction [26], have a limited geographic distribution on the planet [27]. Their vulnerability to environmental changes makes them valuable sentinels of environmental quality [28], helping maintain natural system balance [29,30]. The life of amphibians is largely related to the biology and ecology of many other vertebrates [31,32]. Therefore, the vulnerability of this group affects the complex biodiversity of ecosystems. The latest update from IUCN [8] highlights amphibians as the most threatened group, with 41% of species in a vulnerable state and facing a pronounced rate of decline [1,33]. Global evidence points to negative impacts from invasive species, overexploitation, climate change, excessive ultraviolet radiation, chemical pollutants, and disease spread [34]. Nevertheless, habitat loss is the main process underlying these impacts, for which it has been recognized as the dominant threat to these species, according to various authors [33,35,36].

Chile’s amphibians are not exempt from these threats. The Species Classification Regulation (abbreviated to RCE in Spanish; [37]) and Correa [38] recognize 58 valid amphibian species from 14 genera in Chile. Of them, 22 are classified as “endangered” (EN), 11 as “vulnerable” (VU), and seven as “critically endangered” (CR). As a result, after fish, amphibians comprise the second-highest risk category nationally (71%; [37]). The country’s amphibian richness is concentrated between 38° and 48° S, probably due to the fact that this region also has the highest endemicity [39]. According to MMA [40], the northern region of the nation is home to the most at-risk species. Systematic planning and consideration of intrinsic characteristics of the target species, such as body size-related morphological variables, ecological variables (e.g., degree of speciation), phylogenetic and genetic traits, and environmental variables (e.g., potential threats), are necessary for effective conservation [36,41,42] by means of a methodology that considers various aspects known for these species, with the objective of establishing conservation priorities for those species whose most relevant characteristics can be considered in the conservation plans.

According to Johnson et al. [43], assessments of amphibian prioritization aid in the most efficient use of conservation resources by pointing out actions that would best benefit species that require support. Currently, Chile lacks a species prioritization mechanism, as this decision rests solely with specialists [44]. This research aims to apply a methodology for prioritizing amphibian conservation in Chile by employing a multi-criteria analysis approach that considers various factors or criteria such as geographical locations, conservation status, threat levels, degrees of protection, body size, and taxonomic uniqueness. The results will generate a prioritized list of species that require greater attention and resources for their conservation.

## 2. Materials and Methods

A total of 58 species of amphibians from 14 genera in Chile were evaluated (Table 1). Variations in species numbers were caused by taxonomic disputes within polytypic genera. According to Cuevas & Formas [45], Correa et al. [46], Vidal & Díaz-Páez [36], Lobos [47], Correa et al. [48], Fibla et al. [49], Correa & Méndez [50], Correa [38], and other sources, only species that were well-supported by comprehensive descriptions and updated data were evaluated for this study.

### 2.1. Information Collection

An exhaustive literature review was conducted on the ecology, extent of occurrence, systematics, and conservation categorization processes of amphibian species described in Chile. Additionally, data from citizen science publications (e.g., iNaturalist), species conservation, and governmental plans for 2017–2024 were included [51]. Geographic distribution maps were created from the occurrence records to assess the conservation priority of different species based on specific criteria levels (see below). Other biological factors were also taken, including genetic structure, conservation threats, dispersal and/or reproduction characteristics, and occupancy estimates (e.g., [47,52,53,54]).

### 2.2. Criteria for Prioritization

The function developed from criteria put forth by different authors served as the foundation for the prioritization plan for amphibian conservation and acted as a guide for the condition of each species’ population [7,54,55,56,57]. Due to Chile being considered a biogeographic island within the Neotropical region, not all criteria used in other countries fit the country, showing a limitation in the method. For example, Chile’s biodiversity is very low, but the country is considered a biodiversity hotspot because of its high endemism. Therefore, the use of criteria intended to describe megadiverse countries cannot be applied to Chilean biodiversity [20]. The five dimensions that were used in the prioritization process had nine quantitative assessment criteria each: distribution (which included geographic distribution, habitat breadth, area of occupation, and extent of occurrence); population projection (which included threat level and population trend); evolutionary history (which included taxonomic uniqueness); human impact (which included body size); and land management (which excluded land). Below is a description of these criteria:(a)Geographic distribution (GEDIS): According to Reca et al. [7], the distribution criterion is used to classify the extent of the geographic area that a species occupies. High degrees of endemicity are frequently linked to restricted ranges, which increases their susceptibility to extinction and makes conservation of these species a top priority [22,56]. Reducing biodiversity loss requires knowledge of predetermined spatial scales with equivalent dimensions for the efficient planning of priority conservation areas [58,59]. In this context, if it is expected that the few remaining wilderness areas that support this biodiversity will be effectively protected, it is critical to evaluate the effects of this biotic erosion (i.e., biodiversity loss resulting from human intention in the world’s ecosystems) on various aspects of diversity and overall ecosystem properties [60,61]. Limited geographic distribution has been identified as a conservation constraint for amphibians in Chile, as species with restricted areas are more vulnerable to extinction [36,62].(b)Breadth of habitat (BREHA): According to Úbeda & Grigera [63], this criterion takes the species’ ability to adapt to a variety of habitats into account. Because they are specialist species and more vulnerable to local extinctions when their habitats are disturbed, species with a restricted breadth of habitat indicate a lack of ecological flexibility [17,55,64].(c)Area of occupancy (AOO): A standard measurement in square kilometers of the area occupied by a particular type of ecosystem. AOO (IUCN subcriterion B2) counts occupied grid cells to determine how risk is distributed among occupied patches [65]. An increased likelihood of a potential threat impacting a substantial portion of a taxon’s distribution increases when it inhabits a highly restricted area, thereby increasing the taxon’s risk of extinction [66,67].(d)Extent of occurrence (EOO): A standardized measurement of the region that contains every instance of a particular type of ecosystem. EOO (IUCN subcriterion B1) assesses the spread of risk over a contiguous area that encloses all occurrences using a minimal convex polygon. It represents the area encompassed by imaginary boundaries enclosing all known, inferred, or projected sites of a taxon’s presence [66]. This parameter quantifies potential risk factors that extend across the species’ geographic distribution [67].(e)Level of threat (LET): The threat status of a species is determined based on quantifiable parameters related to its distribution area and population biology [8]. This criterion involves the national categorization of a taxon. In Chile, species are classified according to the RCE established under Decree No. 29 of 2011 by the Ministry of the Environment. The conservation status is classified as follows: Extinct (EX), Extinct in the Wild (EW), Critically Endangered (CR), Endangered (EN), Vulnerable (VU), Near Threatened (NT), Least Concern (LC), Data Deficient (DD), and Not Evaluated (NE).(f)Population trend (POPT): This criterion aims to estimate the population status over time by evaluating changes in the number of individuals in natural populations due to mortality, birth, migration, or emigration [57]. Population decline is measured by the reduction in the number of mature individuals expressed as a percentage over a specific period [56,58]. Populations facing threats are likely to experience significant declines in the number of individuals. This perspective acknowledges that the population size is influenced by various environmental factors affecting the species [57].(g)Taxonomic uniqueness (TAXU): The extinction of a taxon from a polytypic genus is considered less significant than the extinction of a single species within its genus and/or family [68]. Therefore, monotypic species require greater conservation efforts to preserve their unique DNA sequences [7]. Conservation of a species involves understanding its taxonomic identity and biogeographic distribution, both linked to the geological trajectory of the phylogenetic group it belongs to [69]. Preserving phylogenetic diversity in species clades most susceptible to regional or local threats is also emphasized [70]. An example of taxonomic uniqueness is *Calyptocephalella gayi*, a Gondwanan-origin species unique to Chile, facing diverse conservation threats [71].(h)Body size (BOSI): Species with larger body sizes appear to be more vulnerable to extinction risks than small-bodied species, as they can be affected by human hunting and invasive predators due to their higher visibility in natural environments [72,73,74]. Larger species are also more susceptible to negative effects from human phobias and can be objects of trade for purposes like pets, food, or medicinal use [75]. In the case of *C. gayi*, the largest frog in Chile weighing over 1.5 kg, human consumption poses a significant threat [36,55].(i)Land protection (LAND): This variable assesses the territorial protection of a certain percentage of the studied population [56]. Both governmental and private protected areas are crucial for safeguarding potentially threatened amphibians [76]. However, such assessments may not provide an accurate representation of the coverage, feasibility, and effectiveness of protected areas at the country or regional scale [58]. For instance, Vidal et al. [77] found that only 60% of species richness and 30% of endemic species are protected within national parks in Chile, and these values decrease further when considering only amphibians.

### 2.3. Prioritization Function

Five dimensions with nine criteria were assessed on a scale from 1 to 4, based on previous proposals and new ones (Table 2). GEDIS, BREHA, AOO, EOO, LET, and BOSI have values from 1 to 3, while POPT, TAXU, and LAND have values from 1 to 4. To account for the variability of each criterion and ensure equal importance, a standardization Function (1) was evaluated before calculating Conservation Priority (CP).

(1)$$C_{ij}=\frac{2}{C_{M_{j}}-1}\left({Ce}_{ij}-1\right)+1,$$
where:

$C_{ij}$ represents the value that the standardized criterion *j* has for species *i*,

${C_{M}}_{j}$ is the maximum value that criterion *j* can reach,

and ${Ce}_{ij}$ represents the effective value that criterion *j* has for species *i*.

The calculation of Conservation Priority (CP) for each species *i* is estimated through the value *Cij*, which is obtained by summing the standardized criteria of GEDIS, BREHA, AOO, EOO, LET, POPT, TAXU, and BOSI, and then dividing the result by the LAND criterion (Function (2)).
(2)$$CP=\sum_{j=1}^{n}\frac{C_{ij}}{{{nce}_{ij}LAND}_{ij}},$$
where:

CP stands for Conservation Priority,

n represents the total number of criteria,

$C_{ij}$ represents the value of conservation criterion *j* for species *i*, 

${LAND}_{ij}$ represents the value of the Land Protection criterion *j* for species *i*,

and ${nce}_{ij}$ represents the number of criteria *j* evaluated for species *i*.

Prioritization Categories: For the assessment, four priority categories are proposed: No Priority, Low Priority, Medium Priority, and High Priority. Each category was determined based on the frequency distribution of the Conservation Priority values (Figure 1), using quartile intervals Q_1_, Q_2_, and Q_3_. Species with *p* values greater than or equal to the median are categorized as priorities for conservation [64,78].

## 3. Results

A total of 58 amphibian species were evaluated. Table 3 shows the recorded values for each criterion evaluated by each species, which were later used in the priority calculations using Functions (1) and (2). Criteria with missing values (i.e., criterion value = 0) correspond to species with insufficient background information for analysis. The main criterion with deficient information is POPT, as seen in species like *Alsodes australis*, *A. cantillanensis*, *A. hugoi*, *A. kaweshkari*, *A. monticola*, *A. vittatus*, *Batrachyla nibaldoi*, *C. grandisonae*, *E. altor*, *E. contulmoensis*, *E. septentrionalis*, *R. spinulosa*, *T. halli*, and *T. pefauri*. 

The values of Conservation Priority (CP) ranged from *p* = 0.48 to *p* = 3.0 (Table 4). The median value (*p* = 1.67) divided the species into two main groups: non-priority species (*p* < 1.67) and priority species (*p* ≥ 1.67) (Figure 1). Eight priority families were identified: *Alsodidae*, *Batrachylidae*, *Bufonidae*, *Calyptocephalellidae*, *Ceratophryidae*, *Leptodactylidae*, *Rhinodermatidae*, and *Telmatobiidae*, with 14 species requiring more attention, representing 24% of the species analyzed. Within this group of species, the genus *Telmatobius* was found to have six species of highest priority, while *Alsodes* had five, and *Eupsophus*, *Rhinoderma*, and *Telmatobufo* each had one priority species. The species with the highest priority are *T. fronteriensis*, *T. philippii*, *T. halli*, *A. pehuenche*, *A. tumultuosus*, and *T. chusmisensis* (Table 4).

The delimitation of each priority category was determined using quartiles with intervals (Figure 1) of Q_1_ = 0.10, Q_2_ = 1.67, and Q_3_ = 2.33, resulting in four explicit priority levels: (1) No priority (CP = 0.33 to 0.99) for 22 species, representing 38% of the total species; (2) Low priority (CP = 1.00 to 1.66) including 22 species, equivalent to 38% of the considered species; (3) Medium priority (CP = 1.67 to 2.32) comprising eight species, accounting for 14%; and (4) High priority (CP = 2.33 to 3.00) with six species, representing 10% of the total. When comparing the estimated values of each criterion (Table 4) between the species with the highest CP index (*T. fronteriensis*) and the lowest (*A. australis*), the most significant differences are observed in the distributional dimension criteria (GEDIS, BREHA, AOO, and EOO), population dimension (LET and POPT), and the territorial management dimension (LAND) (Figure 2). In this case, the LAND criterion plays a decisive role in the priority calculation.

When analyzing the priority of each criterion for each genus (Figure 3), the most threatened genera for the GEDIS criterion are *Chaltenobatrachus* and *Telmatobius*. Both genera contain species with microendemic distributions or very isolated localities. For the BREHA criterion, the genus at risk is *Insuetophrynus* because this species is monotypic, known in one type of aquatic environment with few possibilities to expand its ecological niche ranges [26]. In the case of the AOO and EOO criterion, *Insuetophrynus*, *Rhinoderma* and *Telmatobius* were included. These three genera contain species endemic to Chile with restricted areas of distribution [36,77].

For the LET criterion, the genera with the highest value are *Alsodes*, *Atelognathus*, *Calyptocephalella*, *Eupsophus*, *Insuetophrynus*, *Rhinella*, *Rhinoderma*, *Telmatobius*, and *Telmatobufo*, which correspond to genera that contain species categorized as highly threatened by mining, the introduction of exotic species, habitat loss, and emerging diseases such as *Batrachochytrium dendrobatidis* and *Ranavirus* [26]. The genera *Calyptocephalella*, *Hylorina*, *Insuetophrynus*, *Nannophryne*, *Rhinoderma*, and *Telmatobufo* stand out for the POPT criterion whose species contain records of decreasing population trends due to local losses and, in the case of *C. gayi*, human consumption [14]. In the analysis of the Population Projection Dimension, 43% of the genera have a maximum value.

When evaluating TAXU, the priority genera are *Calyptocephalella*, *Chaltenobatrachus*, *Hylorina*, and *Insuetophrynus*, because it corresponds to a monotypic species (Table 2 and Table 3). For the BOSI criterion, only one species corresponding to the monotypic genus *Calyptocephalella* reached a larger body size, which is used for human consumption in spite of being protected [36]. Due to the denominator character of the LAND criterion in Function (2), the genus *Chaltenobatrachus* represents a critical assessment for the evaluation of the Territorial Management dimension. Based on the entire set of parameters that make up the Conservation Priority Index, the genus *Telmatobius* was listed as having the highest priority.

## 4. Discussion

Based on the conservation priority analysis, the species *T. fronteriensis* and *T. philippii* share the highest priority for conservation at the national level (*p* = 2.50). These species show similarity in the quantifiable assessment of their criteria, with limited spatial distribution, highly threatened population projection, and lack of territorial protection areas [8,37]. This aligns with the conservation categories and priority analysis, where high-priority species are characterized by dependence on the distributional range, population projection, and territorial protection [8,37,79,80,81,82]. 

The classification of these species as a priority reflects their current conservation status in Chile. The taxonomic status of some species of the genus was studied by von Tschirnhaus & Correa [83], who established that the species *T. dankoi* and *T. vilamensis* are junior synonyms of *T. halli*. Several factors contribute to the classification of these three species as critically endangered. For example, the species *T. dankoi*, now *T. halli*, was categorized as endangered in 2015, leading to the Conservation and Environmental Education project to protect it from the Loa River. Additionally, in 2019, a rescue reaction was initiated for the last living individuals in the Las Cascadas sector in Calama, which was the only known habitat of this species and was seriously threatened. This prompted a rescue and relocation operation carried out by the Ministry of the Environment, amphibian experts, and the Calama Natural History Museum to transfer the captured individuals to the Quebrada Ojos de Opache (62 specimens) and the National Zoo (14 specimens) [84]. While timely conservation efforts were undertaken for this species, a systematic priority assessment for the rest of the country’s amphibian species is lacking. Both *T. halli* and the two species it was synonymized with are currently listed as being critically endangered. These studies serve as a valuable decision-making tool for the conservation of Chile’s amphibian biodiversity.

Similar situations occur in Perú and Bolivia for *Telmatobius culeus* due to human use. The Lake Titicaca frog is the largest fully aquatic frog in the world and is endemic to this lake [85]. In the Peruvian part, the frogs are harvested and transported to different markets where they are consumed as frog juice or extract, as an exotic dish, as canned food, and as flour, as they are supposed to have medicinal properties. As a consequence, the Denver Zoo has committed, through its conservation program, to establish a program that includes field work, captive management, and environmental education on this species in order to obtain biological, habitat, and socioeconomic information on the inhabitants of the shores of Lake Titicaca and establish the basis for the conservation of the species [86,87]. 

The results reveal that certain species, despite being classified as facing a high level of threat according to the IUCN and RCE categories, have wider distributional ranges and some of their populations are already protected within territorial protection areas. Examples include *A. barrioi*, *A. montanus*, *A. verrucosus*, *E. altor*, *E. insularis*, *T. venustus*, and *P. marmoratum*. On the other hand, species like *I. acarpicus*, *A. vanzolinii*, and *E. contulmoensis*, which are endemic or microendemic and face a high level of threat [88], were categorized as low priority because a portion of their populations (<50%) are already protected within a designated area. Conversely, species like *A. australis*, which are not a priority, have only 25% of their populations residing within a protected area [8,37,39,82,89,90,91] (Figure 4).

The evaluation of conservation priorities for Chilean amphibians utilized adapted and standardized criteria. The highest priority values were observed for species in the *Telmatobius* and *Alsodes* genera, with distributional dimension, population projection, and territorial management criteria being the most influential. These criteria mostly resulted in maximum values. In particular, the LAND criterion played a crucial role in the priority calculation [57]. The establishment of private or governmental protected areas significantly enhances species protection against habitat loss and potential threats to amphibians [76]. However, it is essential to consider that even species within protected areas may still face threats due to incorrect management or inadequate habitat coverage for the species present [78,92,93,94,95].

Establishing conservation categories helps assess the level of threat for species [96]. In Chile, 70% of the analyzed amphibians are categorized under some level of threat by the RCE (Species Classification Regulation), leading to the prioritization of certain species over others [96]. As a result, species classified as CR, EN, and VU (RCE) are not prioritized in the CP calculation, with eight of these species falling into the Medium Priority category, 19 into the Low Priority category, and seven classed as No Priority. Prioritizing species has become a crucial aspect of conservation [97]. Similar to conservation categories, priority species lists employ multivariate analysis using scoring criteria to create profiles for each species. This approach facilitates grouping species with similar threat or priority profiles, such as *T. halli*, *T. fronteriensis*, *T. philippii*, and *T. chusmisensis*, which share comparable management challenges [96].

Categorizing amphibians quantitatively in Chile is challenging due to limited available information, where data on taxa abundance and accurate geographic distribution are often lacking [55]. This scarcity of data presents a major issue for biodiversity conservation in the central-southern hemisphere of America [98]. Monitoring amphibian populations and collecting data on their abundance and decline is essential [26,55]. Addressing data uncertainty is also important, as a species may be misclassified as insufficiently known without justification [78].

Analyzing the conservation priority of species in conjunction with potential extinction threats within a specific geographic area would facilitate appropriate management for biodiversity preservation [99]. Species like *T. chusmisensis*, *T. halli*, *T. fronteriensis*, and *T. philippii* fall into the High Priority category and are found near the Alto Loa National Reserve ([82]; iNaturalist, Figure 4), making it beneficial to consider expanding the reserve’s boundaries to provide simultaneous protection for these species. Strengthening protected areas and improving strategic public policies are essential for ensuring the survival and continuity of amphibians in the national territory [76].

Efforts to protect threatened biodiversity worldwide far surpass the available resources for conservation [22], leading to the necessity of identifying priorities [100]. However, the lack of methodological consensus, particularly regarding the study scale, requires further investigation [17,101]. The conservation priority analysis in Chile revealed that the species *T. halli*, *T. fronteriensis*, *T. philippii*, and *T. chusmisensis* have the highest priority for conservation at the national level. These species share similar characteristics, such as limited spatial distribution, highly threatened population projection, and lack of territorial protection areas. Consequently, they require immediate conservation efforts such as the reconstruction of habitats damaged by mining, the strengthening of regulations related to water protection in the desert, state or private protection areas, etc. The assessment of conservation priority for Chilean amphibians utilized standardized criteria, with the highest priority values observed for species in the *Telmatobius* and *Alsodes* genera. Distributional dimension, population projection, and territorial management criteria predominantly influenced their priority status. Notably, the LAND criterion played a crucial role in the priority calculation, as the creation of protected areas significantly enhances species protection against habitat loss and potential threats. However, it is essential to address challenges, such as incorrect management or inadequate habitat coverage, that can still pose threats to species within protected areas. 

Conservation categories play a vital role in assessing the level of threat to a species, with 70% of analyzed amphibians in Chile falling into some level of threat according to the RCE. Species categorized as CR and EN (RCE) are not prioritized in the conservation priority calculation, as they are already considered to be at high risk. Monitoring amphibian populations and collecting data on their abundance and declines are essential for addressing data uncertainty, which poses a significant issue for biodiversity conservation in the region. Analyzing conservation priorities in conjunction with potential extinction threats within specific geographic areas can inform appropriate management strategies and the expansion of protected areas to ensure the survival and continuity of amphibians in Chile’s national territory. Strengthening protected areas and improving strategic public policies are crucial steps towards effectively preserving amphibian biodiversity.

## 5. Conclusions

The conservation priority analysis of Chilean Amphibians highlights areas that require immediate attention for conservation efforts and offers insightful information about the status of amphibian species in Chile. We identified the species (Families Alsodidae and Telmatobiidae) most at risk of extinction and prioritized conservation efforts based on multicriteria analysis.

The results of this study offer a framework for ranking conservation actions, which can be used to guide conservation strategies and initiatives in Chile. Our conservation efforts can be most effectively directed toward the most vulnerable species and their habitats, which will guarantee the long-term survival of amphibians in Chile. This study also emphasizes the significance of continuing research and monitoring to better comprehend the threats to these species and guide future conservation efforts.

## Figures and Tables

**Figure 1 biology-13-00169-f001:**
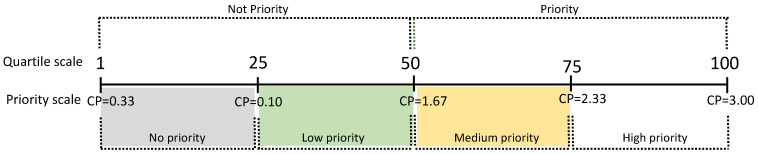
Ranges obtained for Conservation Priority (CP).

**Figure 2 biology-13-00169-f002:**
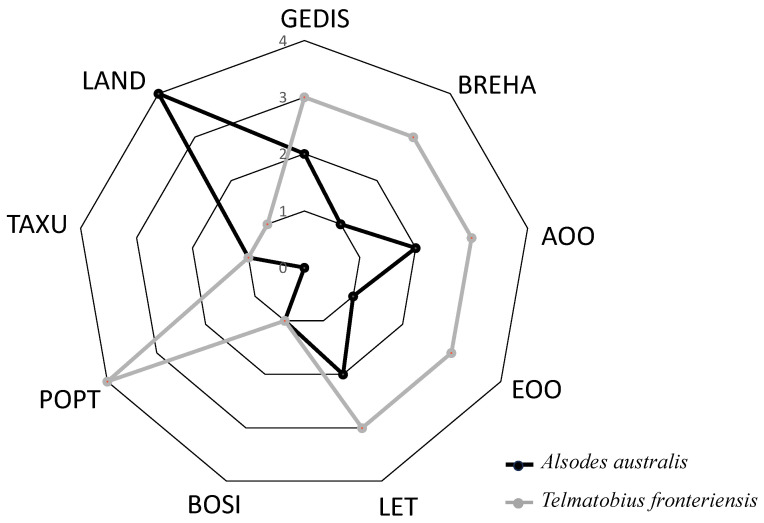
Comparison of evaluation criteria for the species with the highest Conservation Priority value (*Telmatobius fronteriensis*) and the species with the lowest value (*Alsodes australis*). The rating scale for each criterion ranges from 1 to 4, with 1 indicating a low rating and 4 indicating a high rating. Low values of protected land and high values of GEDIS, BREHA, and AOO-EOO characterize the great conservation priority of *Telmatobius fronteriensis*. Geographic distribution (GEDIS), Breadth of habitat (BREHA), Area of occupancy (AOO), Extent of occurrence (EOO), Threat level (LET), Population trend (POPT), Taxonomic uniqueness (TAXU), Body size (BOSI), Land protection (LAND).

**Figure 3 biology-13-00169-f003:**
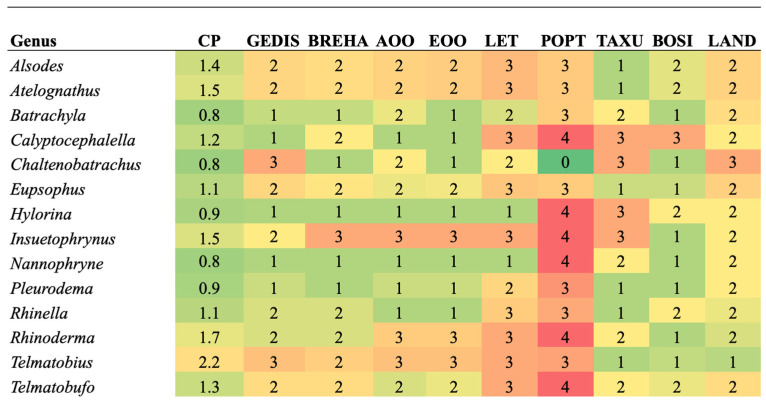
Comparison of criteria for each genus of amphibians evaluated. The color green indicates a low level of threat, and the color red indicates a high level of threat for the criterion. Conservation Priority (CP), Geographic distribution (GEDIS), Breadth of habitat (BREHA), Area of occupancy (AOO), Extent of occurrence (EOO), Threat level (LET), Population trend (POPT), Taxonomic uniqueness (TAXU), Body size (BOSI), Land protection (LAND).

**Figure 4 biology-13-00169-f004:**
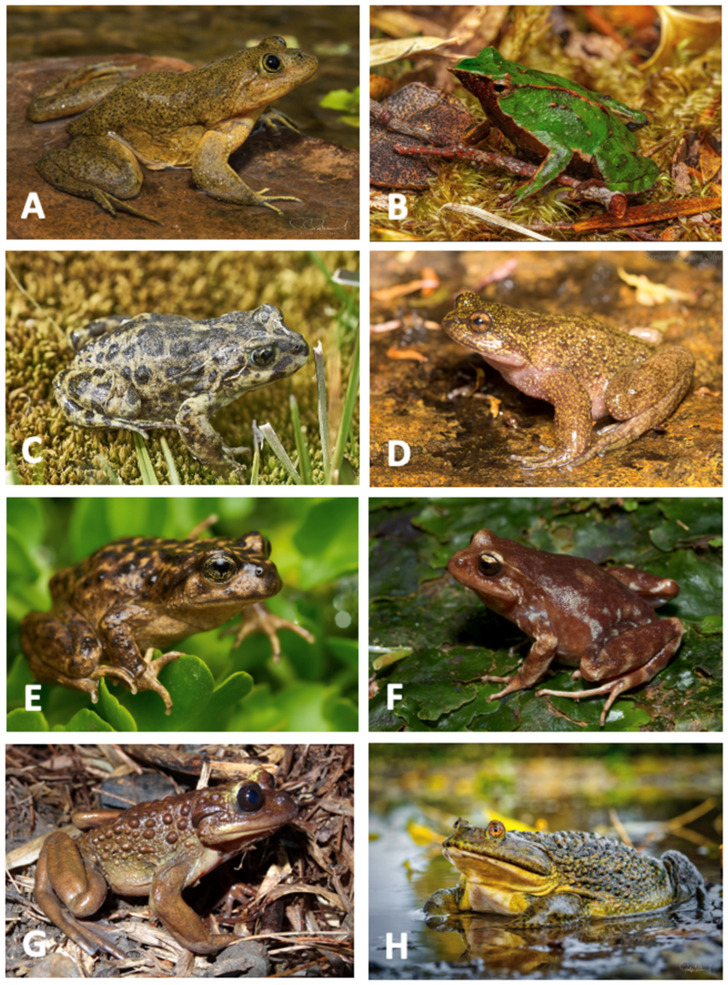
Species that belong to some of Chile’s most endangered genera. (**A**) *Telmatobius chusmisensis*. Photo: Felipe Rabanal, iNaturalist. https://inaturalist.mma.gob.cl/taxa/67227-Telmatobius-chusmisensis (accessed on 22 December 2023); (**B**) *Rhinoderma darwini*. Photo: MatiasG, iNaturalist. https://inaturalist.mma.gob.cl/photos/105185503 (accessed on 22 December 2023); (**C**) *Atelognathus nitoi*. Photo: Daniel Cross, iNaturalist. https://www.inaturalist.org/photos/58533002 (accessed on 22 December 2023); (**D**) *Insuetophrynus acarpicus*. Photo: Bernardo Segura, iNaturalist. https://inaturalist.mma.gob.cl/taxa/23260-Insuetophrynus-acarpicus (accessed on 22 December 2023); (**E**) *Alsodes pehuenche*. Photo: Javier Pérez, iNaturalist. https://inaturalist.mma.gob.cl/taxa/23195-Alsodes-pehuenche (accessed on 22 December 2023); (**F**) *Eupsophus migueli*. Photo: Felipe Rabanal, iNaturalist. https://inaturalist.mma.gob.cl/taxa/23107-Eupsophus-migueli (accessed on 22 December 2023); (**G**) *Telmatobufo bullocki*. Photo: Patrich Cerpa, iNaturalist. https://inaturalist.mma.gob.cl/taxa/23207-Telmatobufo-bullocki (accessed on 22 December 2023); (**H**) *Calyptocephalella gayi*. Photo: Felipe Rabanal, iNaturalist. https://inaturalist.mma.gob.cl/taxa/65148-Calyptocephalella-gayi (accessed on 22 December 2023).

**Table 1 biology-13-00169-t001:** Number of species by genus and family of amphibians present in Chile.

Family	Genus	Number of Species
Alsodidae	*Alsodes*	18
	*Eupsophus*	10
Batrachylidae	*Atelognathus*	1
	*Batrachyla*	4
	*Chaltenobatrachus*	1
	*Hylorina*	1
Bufonidae	*Nannophryne*	1
	*Rhinella*	4
Calyptocephalellidae	*Calyptocephalella*	1
	*Telmatobufo*	4
Leptodactylidae	*Pleurodema*	3
Rhinodermatidae	*Insuetophrynus*	1
	*Rhinoderma*	2
Telmatobiidae	*Telmatobius*	7

**Table 2 biology-13-00169-t002:** Quantitative assessment of the priority criteria for amphibian species in Chile.

Dimension	Criterion	Value 1	Value 2	Value 3	Value 4
Distribution	Geographic distribution (GEDIS)	Four or more administrative regions	Two or three administrative regions	Only present in an administrative region	-
	Breadth of habitat (BREHA)	Chile and neighboring countries	Endemic	Restricted or microendemic	-
	Area of occupation (AOO)	<2000 km^2^	<500 km^2^	<10 km^2^	-
	Extension of presence (EOO)	<20,000 km^2^	<5000 km^2^	<100 km^2^	-
Population trend	Level of threat (LET)	Least Concern	Rare, Insufficiently Known, Data Deficient.	Critically Endangered, Endangered, Vulnerable	-
	Population trend (POPT)	Growing populations in recent years	Populations in recovery or restoration	Populations without significant decline in recent years	Populations declining in recent years
Evolutionary history	Taxonomic uniqueness (TAXU)	Taxon belonging to a genus of more than four species	Taxon belonging to a genus with four or fewer species	Taxon of a monotypic genus	Taxon of a monotypic family
Human effect	Body size (BOSI)	Less than 59 mm	60–129 mm	Greater than 130 mm	-
Land management	Land protection (LAND)	The entire population is outside protected areas	Less than 50% of the population is inside protected areas	50% of the population or more is inside protected areas	The entire population is within protected areas

**Table 3 biology-13-00169-t003:** Criteria evaluated for the 58 amphibian species in Chile. Geographic distribution (GEDIS), Breadth of habitat (BREHA), Area of occupancy (AOO), Extent of occurrence (EOO), Threat level (LET), Population trend (POPT), Taxonomic uniqueness (TAXU), Body size (BOSI), Land protection (LAND).

Dimension	Distribution				Population Trend		Evolutionary History	Human Effect	Land Management
Species	GEDIS	BREHA	AOO	EOO	LET	POPT	TAXU	BOSI	LAND
*Alsodes australis* Formas, Úbeda, Cuevas & Núñez, 1997	2	1	2	1	2	0	1	1	4
*Alsodes barrioi* Veloso, Díaz, Iturra & Penna, 1981	2	3	2	2	3	4	1	2	2
*Alsodes cantillanensis* Charrier, Correa, Castro & Mendez, 2015	3	3	2	2	3	0	1	1	1
*Alsodes coppingeri* Günther, 1881	2	1	2	2	2	3	1	2	2
*Alsodes gargola* Gallardo, 1970	3	1	3	3	3	3	1	2	1
*Alsodes hugoi* Cuevas & Formas, 2001	3	3	2	3	3	0	1	2	4
*Alsodes igneus* Cuevas & Formas, 2005	2	2	2	3	3	4	1	2	4
*Alsodes kaweshkari* Formas, Cuevas & Nuñez 1998	3	3	3	3	2	0	1	2	4
*Alsodes montanus* s (Philippi, 1902)	2	3	3	3	3	4	1	2	3
*Alsodes monticola* Bell, 1843	2	1	2	3	2	0	1	1	2
*Alsodes nodosus* (Duméril and Bibron, 1841)	1	2	1	1	2	4	1	2	2
*Alsodes norae* Cuevas, 2008	3	3	3	3	3	4	1	2	4
*Alsodes pehuenche* (Cei, 1976)	3	1	3	3	3	4	1	2	1
*Alsodes tumultuosus* Veloso, Iturra, y Galleguillos, 1979	2	2	3	3	3	4	1	2	1
*Alsodes valdiviensis* Formas, Cuevas & Brieva, 2002	2	3	2	2	3	4	1	2	3
*Alsodes vanzolinii* (Donoso-Barros, 1974)	2	3	3	3	3	4	1	1	2
*Alsodes verrucosus* (Philippi, 1902)	2	1	2	2	3	4	1	1	2
*Alsodes vittatus* (Philippi, 1902)	3	3	3	2	3	0	1	1	1
*Atelognathus nitoi* Barrio, 1973	2	1	2	2	3	3	1	2	2
*Batrachyla antartandica* Barrio 1967	1	1	2	1	1	3	2	1	3
*Batrachyla leptopus* Bell, 1843	1	1	2	1	1	3	2	1	2
*Batrachyla nibaldoi* Formas, 1997	2	2	2	1	2	0	2	1	2
*Batrachyla taeniata* (Girard, 1855)	1	1	1	1	2	4	2	1	2
*Calyptocephalella gayi* Duméril & Bibron, 1841	1	2	1	1	3	4	3	3	2
*Chaltenobatrachus grandisonae* Basso, Úbeda, Bunge & Martinazzo, 2011	3	1	2	1	2	0	3	1	3
*Eupsophus altor* Núñez, Rabanal & Formas, 2012	3	3	2	2	3	0	1	1	2
*Eupsophus calcaratus* (Günther, 1881)	1	1	2	3	1	3	1	1	2
*Eupsophus contulmoensis* Ortiz, Ibarra & Formas, 1989	2	3	1	2	3	0	1	1	2
*Eupsophus emiliopugini* Formas 1989	2	1	2	1	1	4	1	2	3
*Eupsophus insularis* (Philippi, 1902)	3	3	2	3	3	4	1	1	3
*Eupsophus migueli* Formas, 1978	3	3	2	2	3	4	1	1	1
*Eupsophus nahuelbutensis* Ortiz & Ibarra, 1992	3	2	2	2	3	3	1	1	3
*Eupsophus roseus* (Dumeril & Bibron, 1841)	2	2	1	1	3	4	1	1	2
*Eupsophus septentrionalis* Ibarra-Vidal, Ortiz & Torres-Pérez, 2004	3	3	2	3	3	0	1	1	4
*Eupsophus vertebralis* Grandison, 1961	2	1	2	1	3	4	1	2	2
*Hylorina sylvatica* Bell, 1843	1	1	1	1	1	4	3	2	2
*Insuetophrynus acarpicus* Barrio, 1970	2	3	3	3	3	4	3	1	2
*Nannophryne variegata* (Günther, 1870)	1	1	1	1	1	4	2	1	2
*Pleurodema bufoninum* Bell, 1843	1	1	1	1	2	3	1	1	2
*Pleurodema marmoratum* (Duméril & Bibron, 1841)	2	1	2	2	3	4	1	1	2
*Pleurodema thaul* Lesson, 1827	1	1	1	1	2	3	1	1	2
*Rhinella arunco* (Molina, 1782)	1	2	1	1	3	4	1	2	2
*Rhinella atacamensis* Cei, 1962	2	2	1	1	3	4	1	2	2
*Rhinella rubropunctata* (Guichenot, 1848)	2	1	1	1	3	4	1	2	2
*Rhinella spinulosa* Wiegmann, 1834	1	1	1	1	1	0	1	2	1
*Rhinoderma darwinii* Duméril & Bibron, 1841	1	1	2	3	3	4	2	1	2
*Rhinoderma rufum* (Philippi, 1892)	1	1	1	2	3	4	2	1	1
*Telmatobius chusmisensis* Formas, Cuevas & Núñez, 2006.	2	2	3	3	3	4	1	2	1
*Telmatobius fronteriensis* Benavides, Ortiz & Formas, 2002	3	3	3	3	3	4	1	1	1
*Telmatobius halli* Noble, 1938	3	3	3	3	3	0	1	1	1
*Telmatobius marmoratus* (Duméril & Bibron, 1841)	2	1	2	2	3	4	1	2	2
*Telmatobius pefauri* Veloso & Trueb, 1976	3	3	2	2	3	0	1	1	1
*Telmatobius peruvianus* Wiegmann, 1835	2	1	2	2	3	4	1	1	1
*Telmatobius philippii* Cuevas & Formas, 2002	3	3	3	3	3	4	1	1	1
*Telmatobufo australis* Formas, 1972	2	2	1	1	3	4	2	2	2
*Telmatobufo bullocki* Schmidt, 1952	2	2	1	1	3	4	2	2	1
*Telmatobufo ignotus* Cuevas, 2010	3	3	2	3	3	4	2	2	4
*Telmatobufo venustus* (Philippi, 1899)	2	2	2	2	3	4	2	1	2

**Table 4 biology-13-00169-t004:** List of amphibian species in Chile ordered by level of conservation priority. See Figure 1 for comparison.

Number	Family	Species	Conservation Priority	Type Priority
1	Telmatobiidae	*Telmatobius fronteriensis*	2.50	High
2	Telmatobiidae	*Telmatobius philippii*	2.50	High
3	Telmatobiidae	*Telmatobius halli*	2.43	High
4	Alsodidae	*Alsodes pehuenche*	2.38	High
5	Alsodidae	*Alsodes tumultuosus*	2.38	High
6	Telmatobiidae	*Telmatobius chusmisensis*	2.38	High
1	Alsodidae	*Alsodes gargola*	2.29	Medium
2	Alsodidae	*Alsodes vittatus*	2.29	Medium
3	Alsodidae	*Eupsophus migueli*	2.25	Medium
4	Rhinodermatidae	*Rhinoderma rufum*	2.21	Medium
5	Alsodidae	*Alsodes cantillanensis*	2.14	Medium
6	Ceratophryidae	*Telmatobius pefauri*	2.14	Medium
7	Calyptocephalellidae	*Telmatobufo bullocki*	1.96	Medium
8	Ceratophryidae	*Telmatobius peruvianus*	1.88	Medium
1	Rhinodermatidae	*Insuetophrynus acarpicus*	1.53	Low
2	Alsodidae	*Alsodes vanzolinii*	1.43	Low
3	Alsodidae	*Alsodes barrioi*	1.35	Low
4	Alsodidae	*Eupsophus altor*	1.29	Low
5	Calyptocephalellidae	*Telmatobufo venustus*	1.25	Low
6	Calyptocephalellidae	*Calyptocephalella gayi*	1.23	Low
7	Telmatobiidae	*Telmatobius marmoratus*	1.20	Low
8	Rhinodermatidae	*Rhinoderma darwinii*	1.18	Low
9	Calyptocephalellidae	*Telmatobufo australis*	1.18	Low
10	Bufonidae	*Rhinella spinulosa*	1.14	Low
11	Alsodidae	*Alsodes verrucosus*	1.13	Low
12	Alsodidae	*Eupsophus vertebralis*	1.13	Low
13	Leptodactylidae	*Pleurodema marmoratum*	1.13	Low
14	Bufonidae	*Rhinella atacamensis*	1.13	Low
15	Alsodidae	*Eupsophus contulmoensis*	1.11	Low
16	Alsodidae	*Alsodes montanus*	1.07	Low
17	Alsodidae	*Eupsophus roseus*	1.05	Low
18	Bufonidae	*Rhinella arunco*	1.05	Low
19	Bufonidae	*Rhinella rubropunctata*	1.05	Low
20	Alsodidae	*Alsodes monticola*	1.03	Low
21	Alsodidae	*Eupsophus insularis*	1.02	Low
22	Batrachylidae	*Batrachyla nibaldoi*	1.00	Low
1	Alsodidae	*Alsodes nodosus*	0.98	Non-priority
2	Alsodidae	*Alsodes valdiviensis*	0.96	Non-priority
3	Alsodidae	*Alsodes coppingeri*	0.94	Non-priority
4	Alsodidae	*Eupsophus calcaratus*	0.93	Non-priority
5	Batrachylidae	*Hylorina sylvatica*	0.93	Non-priority
6	Alsodidae	*Alsodes norae*	0.88	Non-priority
7	Batrachylidae	*Batrachyla taeniata*	0.88	Non-priority
8	Alsodidae	*Eupsophus nahuelbutensis*	0.88	Non-priority
9	Calyptocephalellidae	*Telmatobufo ignotus*	0.86	Non-priority
10	Batrachylidae	*Batrachyla leptopus*	0.83	Non-priority
11	Alsodidae	*Alsodes hugoi*	0.81	Non-priority
12	Alsodidae	*Alsodes kaweshkari*	0.81	Non-priority
13	Batrachylidae	*Atelognathus nitoi*	0.80	Non-priority
14	Bufonidae	*Nannophryne variegata*	0.80	Non-priority
15	Leptodactylidae	*Pleurodema bufoninum*	0.78	Non-priority
16	Leptodactylidae	*Pleurodema thaul*	0.78	Non-priority
17	Alsodidae	*Eupsophus septentrionalis*	0.76	Non-priority
18	Batrachylidae	*Chaltenobatrachus grandisonae*	0.76	Non-priority
19	Alsodidae	*Alsodes igneus*	0.75	Non-priority
20	Alsodidae	*Eupsophus emiliopugini*	0.70	Non-priority
21	Batrachylidae	*Batrachyla antartandica*	0.59	Non-priority
22	Alsodidae	*Alsodes australis*	0.48	Non-priority

## Data Availability

All data generated by this study are available in this manuscript.

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
