# Peer review of "Identifying Strategies for Effective Biodiversity Preservation and Species Status of Chilean Amphibians"

_biology, 2024, doi:10.3390/biology13030169_

Round 1

Reviewer 1 Report

Comments and Suggestions for Authors

Dear Authors,

please revise the manuscript according to the comments and recommendations. Answer the questions based on the text.

Reviewer 2 Report

Comments and Suggestions for Authors

In the paper “Conservation Priority Analysis of Chilean Amphibians: Identifying Strategies for Effective Biodiversity Preservation” the author evaluated 58 amphibian species in Chile to determine their conservation priority. This manuscript is well organized, and the drawn conclusions are coherent with the obtained results. Although I have loved reading your work, there are a few grammar issues that I have seen, thus I believe that the text needs to be edited by a native English speaker. I hope to provide very useful suggestions to improve the overall clarity of your study as well as the quality of your analysis. I think that my suggestions look feasible to you, and I believe you will be able to address them. Thus, please take care to do a full revision of your manuscript according to all my comments. Improvements based on my comments will be crucial for acceptance. I have some concerns and suggestions for each aspect of the manuscript. Please see below.

Abstract: I would like to suggest giving more emphasis to the results.

Introduction: The paper is technically sound and the claims are convincing. However I think that some references should be updated. Please, note that the hypothesis and the predictions are unclear, you need to well explain them.

Lines 43 - 46: I think that you should add this important reference to support your sentence: “. In this context, if it is expected that the few remaining wilderness areas that support this biodiversity will be effectively protected, it is critical to evaluate the effects of this biotic erosion on various aspects of diversity and overall ecosystem properties”. I would like to suggest:

Bosso, L., et al., (2024). Integrating citizen science and spatial ecology to inform management and conservation of the Italian seahorses. Ecological Informatics, 79, 102402.

Barnes, R. S. K., et al., (2024). Differential sampling in the assessment of conservation and biodiversity merit: a comparison of the seagrass macrofauna in three nearby South African estuaries. Biodiversity and Conservation, 1-24.

Lines 101 – 109: Please, explain in detail your hypothesis and predictions. You need to expand this sections if you would want to express exactly what you want to do.

Materials and methods: In general, the methods are appropriate and the study seems well conducted, although some details deserve a bit more attention i.e., especially about the methodology and the data.

Lines 142 – 143: I think that you should add this important reference to support your sentence: “. In this context, if it is expected that the few remaining wilderness areas that support this biodiversity will be effectively protected, it is critical to evaluate the effects of this biotic erosion on various aspects of diversity and overall ecosystem properties”. I would like to suggest:

Salinas-Ramos, V. B., et al., (2021). Artificial illumination influences niche segregation in bats. Environmental Pollution, 284, 117187.

Gann, G. D., et al., (2019). International principles and standards for the practice of ecological restoration. Restoration Ecology, 27(S1), S1-S46.

Results: Well written! The figures and the tables are all informative and necessary, but not redundant, ensuring the correct comprehension of the manuscript.

Discussion: The paper discussed appropriately the context and the theme, although there is important literature not cited by the authors. I think that the authors should be discussing their results also comparing them with those already published on other species/genus/family.

Comments on the Quality of English Language

Moderate editing of English language required

Reviewer 3 Report

Comments and Suggestions for Authors

Dear authors,

Congratulations for your work.

Please find below some suggestions:

Introduction

38-46

The citations and references should be much richer and diversified in all the paper! Sometimes the citations of some not original for the paper texts are completely missing.

The causes of species extinction are more numerous then the ones highlighted by you in introduction.

In this context, if it is expected that the few remaining wilderness areas 43 that support this biodiversity will be effectively protected 

It is desired/needed not at all expected in the context in which the human impact have increasing trends.

Define biotic erosion. What type of diversity?

69-79

Locally, Chile has a distinct biodiversity legacy .....

Locally or regionally?

129-127

A phrase in which you argue why you select only these criteria and no others too.

Material and Methods

Please reveal the method limitation.

Discussion

Please reveal potential bridges among your Chilean approach and the adjacent and continental areas in terms of assessment, monitoring and management.

Conclusions

The conclusions should be in my opinion more applicated on the original results.

I hope that these suggestions can help.

All the best

Reviewer

Round 2

Reviewer 1 Report

Comments and Suggestions for Authors Dear authors, I give a positive assessment of the manuscript.